# Hepatitis C Virus Epitope Immunodominance and B Cell Repertoire Diversity

**DOI:** 10.3390/v13060983

**Published:** 2021-05-25

**Authors:** Nicholas A. Brasher, Anurag Adhikari, Andrew R. Lloyd, Nicodemus Tedla, Rowena A. Bull

**Affiliations:** 1Faculty of Medicine, School of Medical Sciences, UNSW Sydney, Sydney, NSW 2052, Australia; nbrasher@kirby.unsw.edu.au (N.A.B.); aadhikari@kirby.unsw.edu.au (A.A.); N.Tedla@unsw.edu.au (N.T.); 2The Kirby Institute, Faculty of Medicine, UNSW Sydney, Sydney, NSW 2052, Australia; a.lloyd@unsw.edu.au; 3Department of Infection and Immunology, Kathmandu Research Institute for Biological Sciences, Lalitpur 44700, Nepal

**Keywords:** hepatitis C virus, human monoclonal antibodies, neutralising antibodies, antigenic domains, epitope mapping, immunodominance, HCV–BCR interactions, public antibody repertoire, vaccine development

## Abstract

Despite the advent of effective, curative treatments for hepatitis C virus (HCV), a preventative vaccine remains essential for the global elimination of HCV. It is now clear that the induction of broadly neutralising antibodies (bNAbs) is essential for the rational design of such a vaccine. This review details the current understanding of epitopes on the HCV envelope, characterising the potency, breadth and immunodominance of antibodies induced against these epitopes, as well as describing the interactions between B-cell receptors and HCV infection, with a particular focus on bNAb heavy and light chain variable gene usage. Additionally, we consider the importance of a public repertoire for antibodies against HCV, compiling current knowledge and suggesting that further research in this area may be critical to the rational design of an effective HCV vaccine.

## 1. Introduction

Globally, it is estimated that over 71 million people are chronically infected with hepatitis C virus (HCV), with an annual disease-specific mortality of approximately 400,000 due to the complications of cirrhosis, liver failure, and hepatocellular carcinoma [1]. While treatment with direct-acting antiviral (DAA) drugs are highly curative, they remain costly, and the health infrastructure to deliver these treatments worldwide is poorly developed. As such, the development of a preventative vaccine against HCV remains an important focus for infectious disease research—a fact that is highlighted in the World Health Organisation’s 15-year hepatitis C elimination plan [2].

Over the past three decades of HCV research, it has become clear that the induction of broadly neutralising antibodies (bNAbs) which target conserved epitopes on the viral envelope is essential for the rational design of an effective HCV vaccine. An ideal vaccine is not likely to confer absolute protection, but would improve the natural clearance rate of 25% (with the remaining 75% developing persistent, chronic infection) [3]. The identification and analysis of the protective capacity of each antibody (Ab) and their specific epitope targets on the viral envelope would allow the prediction of the capacity to reliably clear infection.

Whilst it is important to understand the epitope-binding sites of protective anti-HCV Abs, the B-cell receptor (BCR) characteristics that form the Ab paratope (i.e., the antigen-binding site) should not be overlooked. The characterisation of BCR gene usage, with a particular focus on the variable gene segment and complementarity-determining regions (CDRs), would allow the identification of the distinct Ab features that should be induced by an ideal vaccine to elicit effective, long-lasting protection.

This review will address the current understanding of these viral epitope–BCR interactions and will consider the importance of a public Ab repertoire, which has been investigated in the context of other viruses but insufficiently studied in HCV. Developing a deeper understanding of these elements of the humoral response to HCV will provide key insights that will draw the goal of developing an effective vaccine nearer.

## 2. HCV Envelope Epitopes and Immune Protection

The HCV envelope glycoproteins E1 and E2 are the major targets of the neutralising Ab response, and as such, have been a key focus of research in recent years. In particular, the E2 protein plays a critical role in infection by binding to host entry factors, including CD81, scavenger receptor class B type 1 (SRB1), claudin1, occludin, Niemann-Pick C1-like 1 (NPC1L1), and several receptor tyrosine kinases [4]. Importantly, the binding of E2 to CD81 instigates the process of viral internalisation and, as such, the E2 binding sites for CD81 have been shown to be critical epitopes for neutralising activity [5]. Additionally, the hypervariable regions (HVRs) of E2 feature exceptionally high sequence variability and modulate the affinity and avidity of receptor binding and viral cell entry [6,7]. HVR1 has been of particular interest in recent years due to its proposed role in restricting the binding and activity of neutralising Abs (NAbs), likely by sterically masking key epitopes [8].

Several epitopes have been identified on E1 and E2 by independent groups, resulting in largely inconsistent nomenclature. The epitopes most widely referred to are the antigenic regions (ARs) 1-5, domains A–E and epitopes I–III (Figure 1). Amongst these incongruous epitope definitions lie several which largely overlap with one another, or even represent identical binding residues [9,10,11]. These residues are typically defined by alanine scanning mutagenesis (ASM) studies. Domains D and E, which form part of epitope I and epitope II, respectively, are predominantly linear, whereas most other epitopes that were similarly analysed were comprised of non-linear residues (Table 1, Figure 1) [9,12,13,14,15,16,17].

### 2.1. Variance in the Protection of Antibodies Directed at Distinct Epitopes

The neutralising ability of mAbs generated against these various epitopes varies widely, with some epitopes typically inducing non-neutralising Abs, while others induce broad, strongly neutralising Abs. Most notably, AR3/domain B and AR4 epitopes have been shown repeatedly to induce bNAbs [9,16], whereas Abs against domain A and AR1 Abs have been shown to have limited or no neutralisation ability. Domain C and AR2 often induce Abs with potent neutralisation activity; however, the Abs are often specific to unique viral genotypes [12,27]. Additionally, several studies have looked at the synergistic and interfering effects of two or more NAbs in combination. For example, it has been shown that epitope II-specific Abs can interfere with the neutralisation ability of epitope I-specific Abs [28]. More recent studies have analysed these combinatorial effects of several NAbs, identifying a wide range of synergies and interferences [29,30]. These findings suggest it may be beneficial to have a vaccine that can facilitate the induction of certain Ab specificities over others, with a focus on inducing multiple synergistic Abs with enhanced neutralisation breadth and potency. A contrasting approach was employed in a recent, proof-of-principle murine study, in which a bivalent vaccine was designed based on targeting residues with conserved physicochemical properties in the antigen (i.e., considering glycan and lipid shielding) rather than focusing on genetic variations. It was found that bNAbs were elicited which were active against HCV variants with up to 70% amino acid deviation from the immunogen sequences [31].

Furthermore, in a recent study, an alternate conformation of the E2 neutralising face was identified [32]. It was found that, upon binding by certain NAbs utilising the heavy chain V_H_1-69, which represents one of the most common and most heavily investigated BCR gene classes, the E2 front-layer region became displaced, exposing additional Ab-binding epitopes, which may also be susceptible to neutralisation. More work needs to be done to comprehensively characterise these epitopes and the Abs which target them.

### 2.2. NAbs Targeting Hypervariable Region 1

Hypervariable region 1 (HVR1) lies at the N-terminus of E2 and is incorporated as part of the AR3 epitope, among others. Following the identification of this region of high sequence diversity, its role in NAb recognition has been keenly explored. The development of technologies such as the HCV pseudoparticle (pp) and HCV cell culture (cc) systems allowed for the analysis of the impact of virus lacking HVR1 on NAb recognition. Initially, pseudoparticles with HVR1 removed (ΔHVR1) were found to have an increased susceptibility to NAbs. Studies on humanised mice have suggested that HVR1 may partially mediate Ab resistance and contribute to viral persistence by protecting epitopes associated with neutralisation, conserved across multiple HCV genotypes [33]. These findings provoked a series of further studies characterising escape from NAbs mediated by HVR1, which showed that deleting or mutating HVR1 in HCV pseudoparticles (HCVpps) increased sensitivity to NAbs and made the HCVpps less infectious [34]. Recent studies have considered the use of ΔHVR1 E2 in inducing bNAbs for vaccines; however, conclusions from these studies have not shown clear promise, with poor induction of cross-NAbs despite producing strong cross-binding Abs [35,36]. Conversely, a previous study found that when HVR1, HVR2 and the so-called intergenotypic variable region (igVR, also referred to as HVR3) were all simultaneously removed in a secreted E2 glycoprotein, protein folding and CD81 binding were not affected. However, when only one of these variable regions was deleted, infectivity was greatly reduced if not completely eliminated with a reduction in CD81 binding [37].

### 2.3. Ab Immunodominance

Immunodominance describes the relative abundance of Abs targeting specific epitopes of a given antigen across an antibody-positive (i.e., naturally exposed) population. By characterising the immunodominance of the response to epitopes associated with effective bNAbs, it becomes possible to understand which bNAbs are more readily induced by a wider population, which is critical information in the context of vaccine design. This is a research area that is only just beginning to be explored, with two very recent studies addressing the topic of Ab immunodominance in HCV [30,38]. By defining the unique specificities within the polyclonal Ab responses in patients with acute HCV infection, the relationship between these specificities and the potency and breadth of neutralisation can be resolved. Although different panels of pseudoparticles and reference mAbs were used in these two studies, both reported that strong bNAb responses were enriched with broadly specific bNAbs directed towards AR3, AR4 and domain D. Moreover, there was no particular epitope associated with viral clearance, but rather Kinchen et al. indicated that viral clearance was associated with the presence of multiple NAbs targeting unique neutralising epitopes and reduced activity against non-neutralising domains [30,38]. The other of these two studies also suggested that the immunodominance profile may be influenced by the genotype of the virus (Figure 2) [30]. This suggests that the immunogenic profile of E1/E2 antigens from various viral genotypes should be assessed before selecting vaccine candidates. It should be noted that the induction of bNAbs is also likely to be influenced by the host immune repertoire, as discussed further below.

## 3. The Interplay of HCV Infection and B-Cell Receptors (BCRs)

Historically, the characterisation of the HCV-specific B-cell repertoire was limited to the examination of B cell clones isolated from the liver of patients with chronic HCV infection. Seminal studies in the early 1990s dissected the clonal perturbation of B cells within lymphoid aggregates or follicles, which are a common feature of the histopathology of chronic HCV infection, highlighting the presence of oligoclonal B cells, coining a now widely used term—HCV-driven clonal dominance [39]. Furthermore, early studies reinforced the idea of monoclonal or oligoclonal B-cell expansion in HCV-related chronic liver disease [39,40,41,42,43]. Prior to the single-cell sequencing era, Ab heavy (H) chain variable, diversity and joining (VDJ) gene usage, as determined by polymerase chain reaction (PCR) and Sanger sequencing, was the primary method for B-cell clonal identification. Clonal expansion was indicated by dominant PCR bands, whereas a fully polyclonal pattern was identified by a smear with no specific dominant bands [39]. Its sensitivity depended upon the frequency of B cells present in the sample, often limiting accurate analysis. However, the identification of common B-cell H clonotypes (B cells that are derived from the same VDJ germline genes) highlighted the conserved structural and biochemical properties of the three complementarity-determining regions (CDRs) in antigen recognition [42]. These CDRs (CDR1, CDR2 and CDR3) lie within an Abs V region and are the sites of the greatest variability, determining specific Ab binding. In-depth analyses of the features of the HCV-specific B-cell response and the association with the outcome of primary infection are limited, and studies analysing features of Abs isolated during acute HCV infection are rare. The landscape of B-cell repertoires, namely, specific V and J gene usage and conserved features of CDR2 and CDR3, has been reported by various studies, the majority of which utilised B-cell samples from chronic patients with co-morbidities such as mixed cryoglobulinemia [28] or lymphoma [39,44,45,46,47,48,49].

### Variable Region Immunoglobulin Heavy (V_H_) and Light (V_L_) Chain Gene Usage

Over the last two decades, several studies have explored the idea that there may be a link between the pattern of particular Ab responses and VDJ gene usage against a range of pathogens [50]. It is theoretically predicted that approximately 10^15^ unique Abs can be generated through different VDJ recombination events, but how many of these theoretical clonotypes are present in an individual’s Ab repertoire, particularly in the context of HCV infection, remains unknown [51]. Moreover, how much overlap in these repertoires exists between individuals (i.e., public clonotypes) is yet to be ascertained. In HCV, many of the bNAbs identified to date utilise the V_H_1-69 gene [52] and, encouragingly, have been isolated from patients that have spontaneously cleared single and multiple HCV infection events and only required limited somatic hypermutation (SHM) to achieve neutralising breadth [53,54]. V_H_1-69-derived Abs have also been shown to target a range of distinct epitopes, including domains B, C, D and those epitopes within HVR1 that may contribute to their broad neutralisation despite limited SHM [54].

V_H_1-69 is one of the more dominant gene classes present in the B-cell repertoire, and is commonly induced in viral infection [55]. One study that investigated the HCV-specific B-cell response in patients that cleared the virus and those who developed chronic infection indicated that V_H_1-69 Abs appeared to be at higher frequency in those with chronic infection, suggesting that this repertoire may be utilised only late in the course of chronic viraemia [56]. By contrast, in patients that cleared the virus, other antibodies with bNAb activity were uniquely enriched, specifically V_H_4-39*J4 and V_H_6-1*J6 [56]. Overall, this study identified that V_H_1, V_H_3 and V_H_4 gene families were common both in patients that cleared the virus and in those that developed chronic progressive disease. Another human study that used high-throughput sorting to a linear epitope (E2 residues 483–499) and single-cell sequencing also reported that V_H_1, V_H_3 and V_H_4 were common in HCV infection, but no differences in V_H_ gene usage were observed between disease outcome groups. In addition, this study also reported that V_H_1-69-derived antibodies, as opposed to V_H_1-46-, V_H_3-15- and V_H_3-30-derived antibodies, were associated with higher neutralisation activity [57]. It is encouraging that a recent study using rhesus macaques as a model reported that similar gene usage to the human studies was preferentially expanded following immunisation with E1E2 [58]. These included V_H_1-36, V_H_4-40 and V_H_5-7, which are the human orthologs of V_H_1-69, V_H_4-39 and V_H_5-57, respectively. As in the second human study, in the rhesus macaques the V_H_1 antibodies generally displayed a higher neutralising capacity than the V_H_4 antibodies [58].

Interestingly, V_H_1-69 and V_H_4-59 are frequently expressed in B-cell disorders associated with chronic infection with HCV, namely, mixed cryoglobulinemia and HCV-associated B-cell non-Hodgkin’s lymphomas [45,59]. Patients with HCV-associated type II mixed cryoglobulinemia express Abs mostly encoded by germline V_H_1-69 with biased use of the V_H_1-69/VκA27 gene combination. This gene usage was also found in HCV-associated lymphomas, suggesting that they represent the malignant counterpart of type II mixed cryoglobulinemia [46]. The underlying mechanisms for these associations remain unknown, although it is plausible that they may be a consequence of molecular mimicry or lowering of the immune tolerance threshold due to a sustained hyper-inflamed state [60].

In light of the above, understanding how HCV infection and chronic co-morbidity might affect the generation and maintenance of HCV-specific BCRs is crucial for the rational design of an efficient therapeutic vaccine. Thus, it is imperative to understand the extent of immunoglobulin gene usage and development dynamics of antibody repertoires in order to advance vaccine development.

## 4. An HCV Public Ab Repertoire

Recently, large datasets of shared, or ‘public’, Ab repertoires in responses against different viral infections have been examined to define the typical distribution of Abs, with a particular focus on heavy and light variable gene usage, structure and CDR sequence. These studies include analysis in populations infected with HIV, influenza virus, Ebola virus, malaria, SARS-CoV-2 and HCV [61,62,63,64,65,66,67]. These studies report that the development of bNAbs is often restricted to certain V_H_ gene usage and structure, and the relative abundance of these antibodies within the population and within the individuals’ repertoire influences their induction.

One of the major challenges for HCV vaccine development has been the induction of an immune response with sufficient breadth to combat the large antigenic diversity of the 8 HCV genotypes and 86 subtypes [68,69,70]. Longitudinal studies of natural infection suggest that neutralisation breadth often takes months or years to develop, with some studies indicating that subjects that clear the infection achieve greater breadth earlier [71]. For most bNAbs, key mutations are required to be introduced into the germline to reach breadth [72]. This has led to the concept of germline-targeting vaccines. For this approach to be effective, the targeted germline needs to be common across an ethnically diverse population and be sufficiently represented in the naïve repertoire to ensure it is ‘engaged’ by the vaccine.

A prominent example of this challenge is found in HIV, where it has been shown that, despite the identification of several potent bNAbs, only 10–38% of those infected with HIV-1 are able to generate Abs capable of broad cross-clade neutralisation [73,74]. In addition, many HIV bNAbs have distinctive features, including long CDR3 regions and high levels of SHM. In addition, the unmutated common ancestors often have a low precursor frequency and are not prevalent among subjects, making them difficult to target through vaccination. In HIV, it has been shown that immune tolerance mechanisms select against BCRs with long CDR3s due to their autoreactivity, resulting in a limited frequency in the naïve repertoire, a feature which has also been described for HCV AR3-like bNAbs [75]. Interestingly, in the HCV macaque study, a rapid contraction of the AR3-like V_H_1-36 antibodies (human ortholog V_H_1-69) was observed following vaccination, suggesting that there would be value in the further exploration of human HCV studies [58].

The consensus interpretation of the HIV studies is that aiming for a vaccine which can induce a potent neutralising response may be unrealistic, and that, instead, a more reasonable goal would be to elicit a more moderately neutralising Ab response that is more readily produced among the at-risk population [76]. It has been suggested that focusing vaccine efforts on inducing ‘public’ antibody classes such as VRC01 [77] may be more efficient. VRC01-class antibodies commonly utilise a V_H_1-2*02 heavy chain, with key contacts identified within the CDR H2 and a CDR L3 of 5 amino acids. It is encouraging that a germline-targeting immunogen, eOD-GT8, has shown promise in being able to bind VRC01-class naïve B cells [78].

Similarly, promising work in influenza has indicated that a commonly induced class of neutralising antibodies targeting the haemagglutinin (HA) stem region can often be derived from the germline genes V_H_1-69, V_H_1-18, and V_H_6-1 [79,80]. In malaria, a ‘public’ lineage derived from V_H_3-30 or V_H_3-33 and with potent neutralising activity was identified from multiple donors and indicated a promising pathway for a vaccine [81].

In HCV, AR3-specific antibodies share several structural and biophysical properties; they are preferentially encoded by V_H_1-69 [82] and typically have a conserved hydrophobic pocket in CDRH2 and a long CDR3 loop. The induction of broadly neutralising antibodies encoded by V_H_1-69 has been associated with the natural clearance of HCV [83]. Studies in animal models suggest that E2 antigens can induce AR3-like antibodies with neutralising capacity after minimal SHM and are therefore of great interest [58]. Neutralising AR3-like antibodies with V_H_1-69 usage have also been isolated from clearers and chronic progressors, suggesting that they might be commonly available [18,84], although this has not been closely investigated across ethnic or geographic distributions. However, few studies have performed a detailed assessment of the human B-cell repertoire to HCV to assess how common these antibodies are in the population. Given the preference for longer CDR3s and studies in HIV suggesting that BCRs with long CDR3s may be selected against due to their auto-reactivity and are subsequently often present at lower frequencies in the repertoire, it would be useful to understand the frequency of HCV-specific bNAbs both in the repertoire, and between individuals. This knowledge would assist with understanding which antibodies may be more easily induced by vaccination.

## Figures and Tables

**Figure 1 viruses-13-00983-f001:**
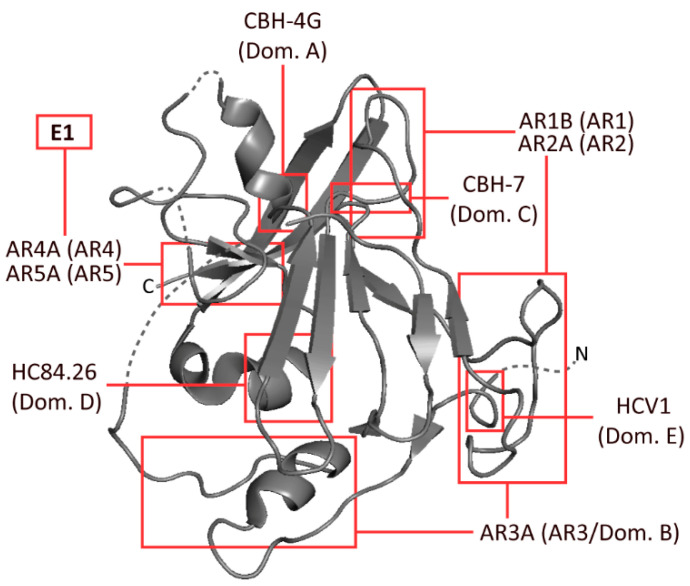
Schematic representation of HCV envelope protein E2, developed from the 2013 crystal structure of the protein by Kong et al. The approximate binding residues of nine mAbs covering a range of epitopes is shown. Structural information taken from the Protein Data Bank—accession code 4MWF [18].

**Figure 2 viruses-13-00983-f002:**
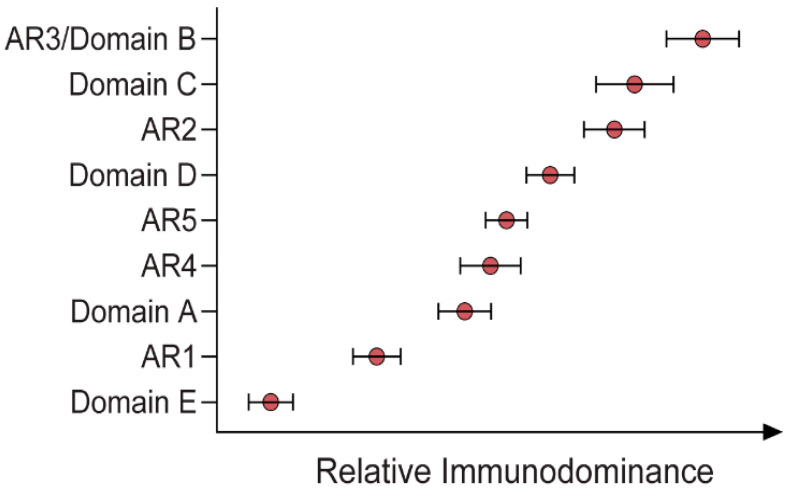
Relative immunodominance of Abs specific to antigenic regions (ARs) 1–5 and domains A–E of a GT1 viral envelope (H77) [30].

**Table 1 viruses-13-00983-t001:** Common human mAbs specific to a range of HCV E2 (and E1) epitopes. Included are antigenic regions (1–5), as defined by Giang et al. [9]; and antigenic domains (A–E), as defined by Keck et al. [10]. Target epitope type was either conformational (C) or linear (L). The binding residues indicate probable binding residues as assessed by alanine scanning mutagenesis (ASM). Neutralisation shown as the half-maximal inhibitory concentration (IC_50_) of H77.

Ab Name	Antigenic Region	Binding Type	Binding Residues (Determined by ASM)	Neutralisation (IC_50_ μg/mL)	References
AR1A	1	C	495, 519, 544, 545, 547, 548, 549, 632	5.7	[12,19]
AR1B	1	C	412, 417, 420–423, 483–489, 523–526, 530–532, 534, 538–540, 544–549	0.06	[12]
AR2A	2	C	625, 628	0.47	[12,19]
AR3A	3/B	C	425, 427–429, 436–438, 440–442, 485, 503, 518, 520, 529, 530, 535, 616	0.5	[12,19]
A8	3/B	C	523, 529, 530, 535	0.56	[20]
CBH-5	3/B	C	523, 525, 530, 535, 540	0.04–13	[21,22]
HC-1	3/B	C	426, 428, 429, 430, 503, 529, 530, 535	0.16	[19,23]
AR4A	4	C	201, 205, 459, 486, 487, 543, 545, 569, 585, 594, 597, 652, 677, 679, 698	0.03–38.5	[9,19]
AR5A	5	C	201, 205, 459, 486, 513, 543, 569, 585, 594, 597, 639, 652, 657, 677, 679	15	[9,19]
CBH-4G	A	C	201, 204–206	>100	[13]
CBH-7	C	C	544, 545, 547, 549, 597, 626	10	[15,19]
HC84.26	D	C	441, 442, 446, 616	0.005–12.91	[16,19]
HC84.27	D	C	425, 426, 428, 429, 441–443, 446, 530, 536, 612, 613, 615	0.22–0.26	[16]
HCV1	E	L	412–423	0.15–15	[14,24]
AP33	E (mouse)	L	412–423	0.6–32	[25]
MAb24	E (mouse)	L	411–428	17.5	[26]

## Data Availability

No new data were created or analysed in this study. Data sharing is not applicable to this article.

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
