# Peer review of "Hepatitis C Virus Epitope Immunodominance and B Cell Repertoire Diversity"

_viruses, 2021, doi:10.3390/v13060983_

Round 1

Reviewer 1 Report

This reviewer welcomes the effort of the authors to present the status of anti-HCV BCR repertoire mapping. The few suggestions are to be found in the text.

Reviewer 2 Report

A very large number of original papers and reviews have been devoted to the topic of broadly neutralizing antibodies (bNAbs) in hepatitis C. This review is interesting because it draws attention to the need to study the "public repertoire" for antibodies against HCV.

Main remarks:

Page 2, fig. 1: Mark domains A, B, C and D to improve schema perception.

Page 3, Table 1: Decipher the abbreviation EC50. Usually, articles on this topic indicate 13-14 main antibodies, while you only indicate 9.

B cell response during acute HCV infection:

I recommend discussing the articles Keck et al. (2019) “Broadly neutralizing antibodies from an individual that naturally cleared multiple hepatitis C virus infections uncover molecular determinants for E2 targeting and vaccine design,” and Rosenberg et al. (2018) “Longitudinal transcriptomic characterization of the immune response to acute hepatitis C virus infection in patients with spontaneous viral clearance”.

Page 11: error in ref. 57

Reviewer 3 Report

Brasher et al. summarized the current knowledge about the humoral response of HCV and its implications on vaccine design. The article is clear and the information is useful to researchers beyond studying HCV. I learned from this article and believe this article will be of interest to many others.

Here are my minor suggestions for consideration:

1) In Figure 1, it would helpful to include the domain names for the indicated antibodies as in Table 1.

2)  Line 68-69, the authors used Epitope I and epitope 2. Whereas in Line 65, the authors used Epitope I-III (Either use Roman or Arabic numeral). It created confusion.

3) Line 199, in Reference 48, macaques were immunized with E1E2 instead of E2. 
